# *Ss*NEP2 Contributes to the Virulence of *Sclerotinia sclerotiorum*

**DOI:** 10.3390/pathogens11040446

**Published:** 2022-04-07

**Authors:** Chenghuizi Yang, Wei Li, Xingchuan Huang, Xianyu Tang, Lei Qin, Yanan Liu, Yunong Xia, Zhihong Peng, Shitou Xia

**Affiliations:** 1Hunan Provincial Key Laboratory of Phytohormones and Growth Development, Hunan Agricultural University, Changsha 410128, China; ychz1995@stu.hunau.edu.cn (C.Y.); weili@cqnu.edu.cn (W.L.); huangxingchuan@stu.hunau.edu.cn (X.H.); chuxuantingnasha@stu.hunau.edu.cn (X.T.); leiqin@stu.hunau.edu.cn (L.Q.); yanan.liu@hunau.edu.cn (Y.L.); yun0623@stu.hunau.edu.cn (Y.X.); 2College of Bioscience and Biotechnology, Hunan Agricultural University, Changsha 410128, China; zhihong_peng@hunau.edu.cn

**Keywords:** *Sclerotinia sclerotiorum*, *Ss*NEP2, ROS, virulence, plant immunity

## Abstract

*Sclerotinia sclerotiorum* is a notorious soilborne fungal pathogen that causes serious economic losses globally. The necrosis and ethylene-inducible peptide 1 (NEP1)-like proteins (NLPs) were previously shown to play an important role in pathogenicity in fungal and oomycete pathogens. Here, we generated *S. sclerotiorum* necrosis and ethylene-inducible peptide 2 (*SsNEP2*) deletion mutant through homologous recombination and found that *Ss*NEP2 contributes to the virulence of *S. sclerotiorum* without affecting the development of mycelia, the formation of appressoria, or the secretion of oxalic acid. Although knocking out *SsNEP2* did not affect fungal sensitivity to oxidative stress, it did lead to decreased accumulation of reactive oxygen species (ROS) in *S. sclerotiorum*. Furthermore, *Ss*nlp24*_SsNEP2_* peptide derived from *SsNEP2* triggered host mitogen-activated protein kinase (MAPK) activation, increased defense marker gene expression, and enhanced resistance to *Hyaloperonospora arabidopsidis* Noco2. Taken together, our data suggest that *Ss*NEP2 is involved in fungal virulence by affecting ROS levels in *S. sclerotiorum*. It can serve as a pathogen-associated molecular pattern (PAMP) and trigger host pattern triggered immunity to promote the necrotrophic lifestyle of *S. sclerotiorum*.

## 1. Introduction

*Sclerotinia sclerotiorum* (Lib.) De Bary is a notorious soilborne ascomycete pathogen that causes stem rot disease [1,2] and can colonize more than 600 plant species, mainly in the Brassicaceae, Solanaceae, Asteraceae, and Fabaceae families [3,4]. *S. sclerotiorum* infects plants by secreting plant cell wall-degrading enzymes (PCWDEs) which soften, hydrolyze, and degrade plant cell tissues [5,6,7,8]. After the plant cell wall is destroyed, oxalic acid (OA) secreted by *S. sclerotiorum* further affects the normal metabolism of host cells. Therefore, the production of oxalic acid is crucial for the virulence of *S. sclerotiorum* [9], while the level of reactive oxygen species (ROS) in plant cells promotes *S. sclerotiorum* infection [10,11]. However, recent studies have shown that OA promotes *S. sclerotiorum* invasion by regulating the pH of the infection environment, rather than the OA itself [12,13].

In a constant struggle with plant immune systems, pathogens have evolved to produce a multitude of effectors, proteins, and small molecules to manipulate the host’s cellular processes and establish a parasitic relationship [14,15,16]. One important family of secreted proteins in pathogenic microorganisms is termed necrosis and ethylene-inducible peptide 1 (NEP1)-like proteins (NLPs) [17]. The first identified member of the NLP family, NEP1 from *Fusarium oxysporum*, causes cell necrosis and induces ethylene production in coca leaves [18,19]. NLPs are highly conserved in bacteria, oomycetes, and fungi [20]. Almost all NLP family members carry an N-terminal secretory signal peptide and a necrosis-inducing Phytophthora protein 1 (NPP1) domain (PF05630) [21]. The predicted signal peptide directs the soluble protein to be secreted into the extracellular matrix. NLPs contain two types of conserved amino acid structures. One is a heptapeptide motif with the sequence GHRHDWE in the central region of the protein, of which -HRH-W- is the conserved core. The other is two highly conserved cysteine residues at the N-terminal half of NLPs [22]. Cytotoxic NLPs bind to glycosyl-inositol phosphoceramide (GIPC) sphingolipids, which are abundant in the outer part of the plant plasma membrane, making NLPs cytolytic in eudicots [22]. The sphingolipids are not just abundant in the outer plasma membrane, they have long hydrophobic lipid chains, they aggregate and form lipid rafts. Most importantly, specific ceramides (GIPC) have been recently defined as NLP protein receptors [23]. For instance, *NLP1* and *NLP2* of *Verticillium dahlia* are cytotoxic, and knockout of *NLP1* and *NLP2* reduced their virulence on tomato (*Solanum lycopersicum*) and *Arabidopsis thaliana* plants [24]. In addition, *VdNLP1* affects vegetative growth and conidiospore production [24]. Similarly, silencing of single cytotoxic *NLP* genes in *Pectobacterium atroseptica* and *P. carotovora* showed reduced virulence on potato tubers, but only the mutant of *P. atroseptica* showed decreased virulence on potato stems [25,26].

NLPs also carry a pattern of 20/24 amino acid residues (nlp20/24), which can be recognized by receptor-like protein 23 (RLP23) in plants [27]. Upon nlp20 induction, pattern-triggered immunity (PTI) responses can be induced, including MAP kinase activation, defense marker gene expression, and enhanced host immunity against virulent pathogens [27,28]. Synthetic nlp24 peptides derived from the oomycete *Hyaloperonospora arabidopsidis NLP3*, nlp24 from *Botrytis cinerea NEP2*, and *NPP1* from *Bacillus subtilis* can also induce ethylene production and trigger plant immunity [29,30]. 

NLPs also have functions beyond plant–pathogen interactions; they affect conidia formation and play an important role in maintaining cell membrane integrity [24,31]. Although NLPs have been studied in different fungal pathogens, their role and molecular mechanism in *S. sclerotiorum* are still unclear. In a previous study, *SsNEP1* and *SsNEP2* from *S. sclerotiorum* were both classified as NLPs, and overexpression of *SsNEPs* in *Nicotiana benthamiana* was reported to cause necrotic lesions on leaves [32]. Here, we discuss the function of *SsNEP2* through genetic knockout and complementation analysis. Our results suggest that *SsNEP2* contributes to the virulence of *S. sclerotiorum* and nlp24 derived from *SsNEP2* could trigger host plant immunity.

## 2. Results

### 2.1. Knockout and Complementation of SsNEP2 in S. sclerotiorum

Although both *Ss*NEP1 and *Ss*NEP2 belong to NLPs [32], the alignment results show that they only share 35% identity in their amino acid sequences (Appendix A). As predicted, both SsNEP1 and *Ss*NEP2 contain N-terminal secretory signal peptide and NPP1 domain (Appendix A). However, only *S**s**NEP2* is abundantly induced upon infection [32], indicating that it may play a more important role in pathogenicity. We replaced *SsNEP2* with the hygromycin-resistance gene in situ by the split marker method to obtain a deletion mutant (Figure 1A). Afterwards, complementation-transformed strains were obtained by means of agrobacterium-induced transformation in *S. sclerotiorum*. PCR results showed that *SsNEP2* was completely deleted and not expressed at all, indicating that Δ*SsNEP2* is a real knockout allele (Figure 1B,C), whereas the complementary strain (*SsNEP2-C*) was restored at the transcription level (Figure 1C). 

To test whether *Ss*NEP2 affects *S. sclerotiorum* growth, we examined both the hyphal and sclerotial phenotypes of the mutant. Δ*SsNEP2* did not affect the mycelium morphology (Figure 2A). The results show that knockout of *SsNEP2* had little effect on sclerotial formation, as there was no significant difference in the number or weight of sclerotia per plate among Δ*SsNEP2*, wild-type, and *SsNEP2-C* strain (Figure 2C–E). Similarly, no significant difference was detected in the mycelium growth rate (Figure 2B), indicating that *Ss*NEP2 does not contribute to the growth of mycelium and the formation of sclerotia. 

### 2.2. Loss of SsNEP2 Compromises Virulence of S. sclerotiorum

Pathogenicity is vital for successful infection by pathogens. To determine whether pathogenicity is affected by *SsNEP2* knockout in *S. sclerotiorum*, mutant and wild-type Δ*SsNEP2* and complementary strain *SsNEP2-C* were inoculated on detached leaves of *A. thaliana* (Figure 3A) and *Nicotiana benthamiana* (Figure 3C). After 24 h, the lesions caused by Δ*SsNEP2* were significantly smaller (<50%) than those caused by the wild-type strain, while the lesions caused by the complementary strain *SsNEP2-C* were similar to those caused by the wild-type strain (Figure 3B,D). After detached leaves were stained with trypan blue, the areas of stained dead cells were consistent with the areas of corresponding lesions of infected leaves. Similar results were obtained for infection on undetached leaves of living plants in *A. thaliana* (Figure 3A) and *N. benthamiana* (Figure 3A,C), indicating that the loss of *Ss*NEP2 compromised the virulence of *S. sclerotiorum*, whereas transformation of the WT copy of *SsNEP2* restored the virulence in Δ*SsNEP2* mutant.

### 2.3. SsNEP2 Does Not Affect Oxalate Secretion and Appressorium Formation

As both oxalic acid content and changes in pH value are very important factors affected the infection process of *S. sclerotiorum* [33], we inoculated Δ*SsNEP2* mutant on bromophenol blue containing PDA for 48 h to check whether the reduced virulence was caused by oxalic acid production. As shown in Figure 4C, Δ*SsNEP2* mutant had the same phenotype as the wild-type and *SsNEP2-C* strains, demonstrating that the mutant secretes oxalate normally and causes a pH change around the hyphae. These experiments suggest that the compromised virulence of Δ*SsNEP2* is not due to a loss of the ability to secrete oxalic acid or to regulate ambient pH.

The appressorium plays an important role in the disease cycle, and reduced appressorium formation leads to decreased virulence of *S. sclerotiorum* [34,35]. To test whether the reduced virulence of Δ*SsNEP2* is due to reduced appressorium formation, we carefully observed the formation and number of appressoria in the mutant. On the slide surface, the Δ*SsNEP2*, wild-type, and *SsNEP2-C* strains formed appressoria normally, and the quantities did not show statistical differences (Figure 4A). After 16 h of infection on onion epidermis, the appressoria were stained with trypan blue, and we found that the mutant could still form appressoria normally during the infection process, which was not significantly different from wild-type and *SsNEP2-C* (Figure 4B). Thus, the reduced virulence of Δ*SsNEP2* is not related to the formation of appressorium.

### 2.4. Knockout of SsNEP2 Affects ROS Accumulation

Cytotoxic NLPs can cause H_2_O_2_ accumulation in plants, leading to elevated ROS levels, which triggers programmed cell death (PCD) [31,36]. As a member of the NLPs, *Ss*NEP2 may also have such a function. *A. thaliana* was infected with WT, Δ*SsNEP2*, and *SsNEP2-C* for 8 h, and then stained with 3,3′-diaminobenzidine (DAB). It was found that significantly fewer brown complexes formed in the leaves of plants infected with Δ*SsNEP2* than the wild type, indicating that *Ss*NEP2 may induce increased H_2_O_2_ content in plants (Appendix A). Pathogens can also secrete ROS, and the developmental differentiation and virulence of fungi are affected by intracellular ROS levels [37,38]. As shown in Figure 5A, the color of Δ*SsNEP2* was lighter than that of the wild type, whereas the color of the complemented strain *SsNEP2-C* was not significantly different, indicating that H_2_O_2_ content in the mutant strains was lower than that in the wild-type strain, and *SsNEP2* knockout did affect ROS accumulation (Figure 5A). 

In addition to acting as effectors, NLPs also affect the growth and development of pathogens and their responses to abiotic stresses [24,31]. To test the function of *Ss*NEP2 in response to peroxide stress, we inoculated WT, Δ*SsNEP2*, and *SsNEP2-C* on PDA medium containing different gradient concentrations of H_2_O_2_. Inoculation of Δ*SsNEP2* and wild type on PDA medium containing H_2_O_2_ cultured for 48 h also showed no difference in the inhibition of mycelial growth at different molar concentrations of H_2_O_2_ among Δ*SsNEP2* mutant, wild-type, and *SsNEP2-C* strains (Figure 5B,C). This suggests that deletion of *Ss*NEP2 does not affect tolerance to H_2_O_2_ of *S. sclerotiorum*.

### 2.5. Ssnlp24 Derived from SsNEP2 Triggers MAPK Activation in Arabidopsis

Just as the conserved 24 amino acids of NLP were reported to activate *Arabidopsis* immunity [27], *SsNEP2* contains the homologous sequence of nlp24 (named *Ss*nlp24*_SsNEP2_*); therefore, we performed multiple alignment of the amino acid sequences of *Ss*NEP2, *Bc*NEP1, *Bc*NEP2, *Bs*NPP1, and *Ha*NLP3. As shown in Figure 6A, these proteins had a high similarity in their amino acid sequences.

To test whether *Ss*nlp24*_SsNEP2_* can trigger plant immunity, we collected *Arabidopsis* leaves after treatment with each synthetic nlp peptide for 10–20 min. From the Western blot results, we found that nlp24 induced MAPK activation, as reported previously (Figure 6B). Notably, *Ss*nlp24*_SsNEP2_* also induced MAPK activation, to the same extent as nlp24 (Figure 6B). These results demonstrate that *Ss*nlp24*_SsNEP2_* induces a PAMP-triggered immune response similarly to nlp24, and the recognition of *Ss*nlp24*_SsNEP2_* in the early infection phase contributes to host immunity against *S. sclerotiorum.*

### 2.6. Ssnlp24_SsNEP2_ Acts as PAMP Signal to Induce Plant Immunity

Treatment with the synthetic nlp20/nlp24 peptide induces *PR* gene expression and ethylene production in *A. thaliana* through the pattern-triggered immunity (PTI) pathway [28,30]. To test whether *Ss*nlp24*_SsNEP2_* also upregulates *PR* gene expression as in conserved nlp24, we took samples pretreated with 1 μM nlp24 or *Ss*nlp24*_SsNEP2_* after 4 or 24 h, respectively. As expected, *Ss*nlp24*_SsNEP2_* enhanced the expression levels of *AtPR1* and *AtPR2* (Figure 7A,B). Over time, the expression levels of both *AtPR1* and *AtPR2* further increased (Figure 7A,B).

To further check whether enhanced *PR* gene expression leads to increased resistance to pathogens, we sprayed spores of *H. arabidopsidis* (*H. a.*) Noco2 on seedlings pretreated with *Ss*nlp24*_SsNEP2_* and found that resistance against *H. a.* Noco2 was enhanced, similar to the nlp24-induced immune response (Figure 7C). These results suggest that *Ss*nlp24*_SsNEP2_* can be recognized as a PAMP and induces plant immunity though PTI.

## 3. Discussion

The virulence of *S. sclerotiorum* is regulated by multiple factors [39]. A key factor is the secretion of proteins out of cells, and the NLP family makes up most of the secreted protein [40]. The results of our assays show that the virulence of the deletion mutant was significantly reduced. The complemented strain, however, showed recovered infectivity at the same level as the wild type. Therefore, *SsNEP2*, as part of a class of cytotoxic NLPs, is related to the virulence of *S. sclerotiorum*. Deletion of this gene results in decreased virulence. *Pya*NLP can bind eudicot GIPCs to exert its cytolytic activity [23]. Whether *Ss*NEP2 has the function of binding to the GIPCs of eudicot requires more in-depth study. However, Δ*SsNEP2* may also secrete OA normally and change the pH value around the hyphae. At the same time, appressoria of the mutant strain can be formed normally and they are no different from that of WT in quantity and quality. Therefore, we demonstrated that the reduced virulence of Δ*SsNEP2* may be due to other reasons that are not related to oxalate secretion or appressorium formation.

In the process of interacting with pathogens, ROS accumulation in plants may occur for two reasons. On the one hand, plants will generate ROS as a molecular signal to improve their own defense status after recognizing pathogens [41]; on the other hand, pathogens can secrete ROS by themselves, which leads to excessive ROS in the plants and triggers programmed cell death [42,43]. After inoculation with the deletion mutant Δ*SsNEP2*, the hydrogen peroxide content in *Arabidopsis* plants was significantly lower compared to those infected with the wild-type strain. *S. sclerotiorum* can manipulate host redox environment to facilitate its infection. For example, by secreting oxalic acid which regulates the production of ROS in plants and leading to PCD [10], or reducing the accumulation of ROS in plants by reducing the production of ROS in *S. sclerotiorum* [44]. Since the mutant does not affect the production of oxalic acid, we speculate that *Ss*NEP2, as a toxic protein, can increase the ROS level in plants and cause cell death, which is beneficial to subsequent *S. sclerotiorum* infection. In addition, we found that *Ss*NEP2 also affected H_2_O_2_ accumulation in *S. sclerotiorum.* Deletion of *SsNEP2* may decrease the content of H_2_O_2_ released by *S. sclerotiorum* into plants, thus reducing the amount of death cell and the damaged area in plants. It is suggested that *Ss*NEP2 may affect virulence by regulating the ROS concentration in the plant and/or *S. sclerotiorum*. The oxygen burst produced by the plant’s immune system also affects the growth of pathogens, which have evolved scavenging systems to maintain the ROS balance in their own bodies [43,45]. We then simulated oxidative stress on *S. sclerotiorum* as it encountered an oxygen burst in plants. However, the performance of Δ*SsNEP2* was not different from the wild type, indicating that deleting *SsNEP2* does not affect the clearance of H_2_O_2_ in *S. sclerotiorum*. 

As one of the earliest signals generated after plants sense pathogens’ invasion, MAPK activation plays an important role in regulating plant immunity [46]*. Ss*nlp24*_SsNEP2_*, which is derived from *Ss*NEP2 contains the homologous sequence of nlp24 and can induce plant immunity such as nlp24. After spraying synthesized small peptides on the surface of plants for 10 min, MAPK cascade was activated. The expression of *AtPR1* and *AtPR2* was upregulated 4 h after induction. Even 24 h later, the expression levels were further increased. This series of reactions is beneficial to improve plants’ resistance to some pathogens. Thus, *Ss*nlp24*_SsNEP2_* treatment increases the resistance to *H.**a.* Noco2. Considering that *S. sclerotiorum* is a necrotic pathogen, the programmed cell death in the plant caused by MAPK activation will benefit its growth. Therefore, we speculate that Ssnlp24*_SsNEP2_*, as a pathogen-associated molecular pattern, can be recognized by plant receptors, thus triggering plant immunity and rapidly increasing ROS levels, resulting in cell damage and PCD, which is more conducive to infection caused by *S. sclerotiorum.*

Previous research has shown that *SsNEP2* was abundantly induced upon infection. After infecting *Arabidopsis* with wild-type strains for 24 h, we measured the expression of *SsNEP2* and found that the expression of *SsNEP2* was significantly up-regulated upon infection (Appendix A). The results were consistent with previous studies, suggesting that *SsNEP2* is involved in the *S. sclerotiorum* infection process. Combined with the protein domain prediction of *Ss*NEP2, we speculate that *Ss*NEP2 may be involved in the infection of *S. sclerotiorum* as a secreted protein. Furthermore, *SsNEP1* expression was increased in Δ*SsNEP2*, either grown on PDA or infected on *Arabidopsis* (Appendix A). We hypothesize that *Ss*NEP1 and *Ss*NEP2 may function redundantly, but this requires more experiments to prove it. Based on the above results, we speculate that *SsNEP2* has an effect on the infection process of *S. sclerotiorum* in two ways. On the one hand, *SsNEP2* is involved in *S. sclerotiorum* virulence as a secreted protein. On the other hand, the small peptide *Ss*nlp24*_SsNEP2_* derived from *Ss*NEP2 triggers plant immunity, induces ROS production, and causes host cell death to promote the necrotrophic life of *S. sclerotiorum*.

## 4. Materials and Methods

### 4.1. Fungal Strains and Culture Conditions

Wild-type strain 1980 was subcultured on potato dextrose agar (PDA), and deletion mutants and complemented strains were cultured on PDA containing 150 μg/mL hygromycin B (Roche) in an incubator at 20 °C for daily storage and 4 °C for long-term storage.

### 4.2. Plant Materials and Growth Conditions

The *A. thaliana* and *N. benthamiana* used for *S. sclerotiorum* virulence assays were cultured in a culture room at 22 °C with 16 h of light and 8 h of darkness; 4-week-old plants were used for the assays. 

### 4.3. Identification and Sequence Analysis of SsNEP2

The NLP sequences of *Fusarium oxysporum Fo47, Verticillium dahliae*, *Magnaporthe oryzae*, *Botrytis cinerea B05.10*, *Aspergillus fumigatus*, *Stagonospora* sp. *SRC1lsM3a*, and *S. sclerotiorum* were downloaded from the Ensembl Fungi database. Multiple sequence alignment of the NLP protein sequences was performed by Clustal software and opened with MEGA7.0 software, then the neighbor-joining (NJ) method was used to construct an evolutionary tree. Bootstrap was set to 1000. Protein sequence numbers are shown in the evolutionary tree. The domains contained in the sequence were analyzed by PredictProtein (https://predictprotein.org/ accessed on 16 February 2022).

### 4.4. Gene Replacement and Complementation

The *SsNEP2* gene was knocked out from the genome of *S. sclerotiorum* using the split-marking approach. Primers 1F: ss12g090490UF (ATCTTCAAAGTTTCCGCCACA)/2R: ss12g090490 UR (GTGCTCCTTCAA-TATCATCTTCTGCAAGAATGGACATCAAGCTC) were designed to amplify approximately 1000 bp of the 5′ untranslated region of *SsNEP2*. Primers 3F: ss12g090490 DF (GTTTAGAGGTAATCCTTCTTTTGGGGTCGAGGTGGAGTGA)/4R: ss12g090490 DR (CAGGAAACCTTGAATGGCTTGA) were used to amplify the 3′ UTR of *SsNEP2* with approximately 1300 bp, using the genomic DNA of *Sclerotinia* as a template. Fragment 1 was composed of the 5’ UTR of *SsNEP2* and 5′ part of hygromycin phosphotransferase cassette, and fragment 2 was composed of the 3’ UTR region of *SsNEP2* and 3′ part of hygromycin phosphotransferase cassette. These two fragments were then inserted into the T-vector (*pEASY*^®^-Blunt Cloning Kit, TransGen, Beijing, China). The resulting vector, T-NEP2 was used as a template to amplify two split-marker fragments using primers ss12g090490 UF (ATCTTCAAAGTTTCCGCCACA)/HY-R (AAATTGCCGTCAACCAAGCTC) and YG-F (TTTCAGCTTCGATGTAGGAGG)/ss12g090490 DR (CAGGAAACCTTGAATGGCTTGA). These two fragments overlapped in the hygromycin resistance gene fragment and were used to co-transform into wild-type *Sclerotinia* protoplasts according to the method of Rollins [47]. ss12g090490 DDF (CGCTCGTCCGTGCATCAAGACT)/HY-R and YG-F/ss12g090490 DDR (GCTGGATTGAAATGCGATGCT) two pairs of primers are used to test whether the hygromycin-resistance gene is inserted at the correct site. Transformants were purified by hyphal tip transfer at least 3 times. Amplified full-length *Ss*NEP2 sequence was used to verify gene deletion strains. 

For Δ*SsNEP2* complementation, the binary vector *pCH-EF-1-NEP2* was constructed using the backbone of *pCH-EF-1* (shared by D. Jiang from Huazhong Agricultural University). The full-length *S**sNEP2* gene, including promoter and coding sequence (CDS), was amplified from WT genomic DNA. Full-length *SsNEP2* gene fragment and *pCH-EF-1* vector were digested by using restriction enzyme *Xho*I and *Sac*I, then linked by homologous recombinase (ClonExpress^®^ II One Step Cloning Kit, Vazyme, Nanjing, China) to generate the *pCH-EF-1-NEP2* construct. Then the plasmid was used for *SsNEP2* transformation using the polyethylene glycol (PEG)-mediated transformation method [47,48].

### 4.5. RT-qPCR Analysis

Gene expression levels in *Arabidopsis* leaves were detected at 4 and 24 h after spraying nlp20 and *Ss*nlp24*_SsNEP2_*. The RNA of test samples was extracted using an Eastep^TM^ Super Total RNA Extraction Kit and GoScript™ Reverse Transcription System Kit for cDNA synthesis (Promega, Madison, WI, USA). Quantitative expression assays were performed using SYBR^®^ Green Premix Pro Taq HS qPCR Kit II (AG11702, Accurate Biotechnology (Hunan)Co.,Ltd, Changsha, China) with StepOne^TM^ Real-time PCR Instrument Thermal Cycling Block. Primers of *AtPR1*, *AtPR2*, and *Actin1* used for qPCR were as follows. PR1-F: (GCTAACTACAACTACGCTGC)/PR1-R: (CTCGTTCACATAATTCCCAC); PR2-F: (CGTTGGAAATGAGGTGAAA)/PR2-R: (CAGTGGTGGTGTCAGTGGC); and ACTIN1-RT-F: (CGATGAAGCTCAATCCAAACGA)/ACTIN1-RT-R: (CAGAGTCGAGCACAATACCG). Actin1 was used as internal reference.

Mycelia of *S. sclerotiorum* were cultured on cellophane and collect *Arabidopsis* leaves which inoculated *S. sclerotiorum* after 24 h for *SsNEP1* and *SsNEP2* gene expression assay. Primers of *SsNEP1*, *SsNEP2* and β-tubulin used for qPCR were: SsNEP1 qF: (GAGGGCAACGACATCCAAG)/*Ss*NEP1 qR: (ACGGCAGCCAGCGGAGATA), *Ss*NEP2 qF: (CCGCCATAGCCATTCCCTTC)/*Ss*NEP2 qR: (ATCACCATTGCTACTGCCACTT) and *Ss*TubqF: (ACCTCCATCCAAGAACTC)/*Ss*TubqR: (GAACTCCATCTCGTCCAT). β-tubulin was used as internal reference. The program setting included holding stage (95 °C, 2 min), cycling stage (95 °C, 20 s; 55 °C, 20 s; 72 °C, 20 s; 40 cycles), and melt curve stage (95 °C, 15 s; 60 °C, 1 min). The transcript level of the gene of interest was calculated from the threshold cycle using the 2^−^^ΔΔCT^ method [49] with three replicates, and the data were analyzed using SPSS Statistics v.24.0 (IBM, Armonk, NY, USA).

### 4.6. Analysis of Virulence

To assess virulence, mycelial plugs (2 or 5 mm in diameter) from wild-type, Δ*SsNEP2*, and *SsNEP2-C* strains were harvested from the edges of colonies maintained in PDA cultures for 48 h and then inoculated on leaves, with 2-mm plugs for *A. thaliana* and 5-mm plugs for *N. benthamiana*. Lesion areas were measured 48 h later and counted with ImageJ. Two leaves were used per experiment and each experiment was repeated 3 times. The data were analyzed using SPSS Statistics v.24.0 (IBM, Armonk, NY, USA).

The infected plant leaves were placed in 1% trypan blue solution and soaked at room temperature for at least 2 h. After that, they were decolorized with 90–100% alcohol and photographed and stored in 60% glycerol.

### 4.7. Compound Appressoria Observation and OA Analysis

A 5-mm mycelia plug of *S. sclerotiorum* was placed on a glass slide and cultured for 16 h to observe the formation and number of appressora. After 16 h of inoculation with S. sclerotiorum, onion epidermis was soaked in 0.5% trypan blue solution for 30 min and then decolorized using bleaching solution (ethanol:acetic acid:glycerol = 3:1:1). Samples were examined and photographed under the light microscope (Axio Imager 2, ZEISS, Oberkochen, Germany).

*S. sclerotiorum* was inoculated on PDA medium containing 100 μg/mL bromophenol blue to detect whether it secreted oxalic acid.

### 4.8. DAB Staining

Using a sterile punch (5 mm), the mycelia-colonized plugs were punched out from the wild-type, Δ*SsNEP2*, and *SsNEP2-C* strains of *S. sclerotiorum* (9 pieces each) and placed separately in a 24-well plate. A total of 2 mL of 1 mg/mL DAB solution was added into each well, and then the samples was incubated for 30 min at 25 °C in the dark and observed immediately.

After 8 h of inoculation with *S. sclerotiorum*, *Arabidopsis* leaves were soaked in 1 mg/mL DAB solution overnight and then decolorized with bleaching solution and photographed (Stemi 508, ZEISS, Oberkochen, Germany).

### 4.9. Pathogen Infection Assay

To analyze nlp20- and *Ss*nlp24*_SsNEP2_*-induced pattern-triggered immunity, leaves of 3-week-old *Arabidopsis* were sprayed with 1 µM nlp20 or Ssnlp24*_Ss_*_NEP2_. After 24 h, the seedlings were sprayed with *H.a.* Noco2 spore suspension at 30,000 spores/mL H_2_O. Plants were then covered with a transparent lid and grown in a short-day incubator at 18 °C for 7 days. Using a cell counting plate, the number of conidia spores per leaf was counted. Experiments were conducted three independent times, and two biological replicates were used each time.

### 4.10. MAPK Activity Assay

Two-week-old *Arabidopsis* seedlings grown on 1/2 MS medium plates were sprayed with 1 µM of nlp20 or *Ss*nlp24*SsNEP2*, and samples were collected 0, 10, 15, and 20 min after each treatment. After grinding in liquid nitrogen, protein extract buffer (200 mM pH = 6.8 Tris-Cl, 8% SDS, 0.4% bromophenol blue, 40% glycerol, 20% β-mercaptoethanol) was added to extract total protein. Phosphorylation of MAPK was detected by Western blot with anti-pERK (no. 4370S, 1:2500 dilution; Cell Signaling Technology, Boston, MA, USA). All experiments were repeated at least 3 times.

### 4.11. Oxidative Stress Response

To test the response of Δ*SsNEP2* to oxidative stress, wild-type, Δ*SsNEP2*, and *SsNEP2-C* strains were grown on PDA medium containing H_2_O_2_ (2.5, 5, 7.5, 10, and 12.5 mM). After 48 h, the diameter and growth inhibition rate of mycelia were measured: Inhibition rate (%) = 100 × (colony diameter of strain without H_2_O_2_—colony diameter of strain with H_2_O_2_)/(colony diameter of strain without H_2_O_2_).

## Figures and Tables

**Figure 1 pathogens-11-00446-f001:**
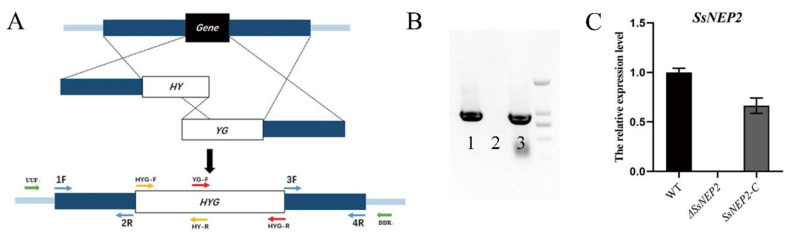
Generation of knockout and complementation strain of *SsNEP2* in *S. sclerotiorum.* (**A**) Split marker method technical process. Fragment 1 is composed of upstream of target gene and the first half of hygromycin-resistance gene (HY), and fragment 2 is composed of the second half of hygromycin (YG) and downstream of the target gene. Both fragments replace the target gene with hygromycin-resistance gene by homologous recombination. (**B**) Full-length *SsNEP2* gene amplification in genomes of WT (1), Δ*SsNEP2* (2), and *SsNEP2-C* (3). (**C**) Relative expression level of *SsNEP2*. β-tubulin was used as internal reference. Relative expression of *SsNEP2* in WT strain was set as control. WT, wild type; Δ*SsNEP2*, knockout strain; *SsNEP2-C*, complementation strain.

**Figure 2 pathogens-11-00446-f002:**
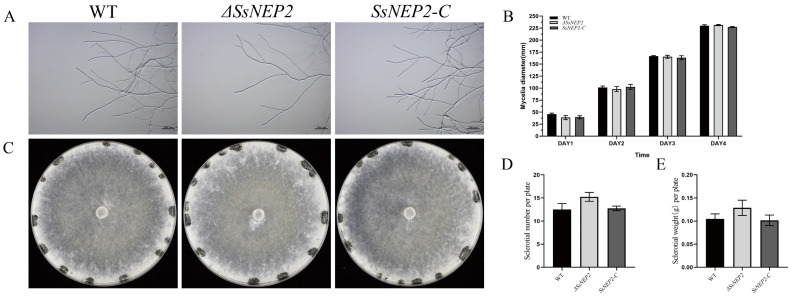
*SsNEP2* deletion does not affect growth phenotype. (**A**) Mycelial morphology of *SsNEP2* knockout and complementation strain. Bar = 0.5 mm. (**B**) Mycelial growth of WT, Δ*SsNEP2*, and *SsNEP2-C*. Strains were grown on PDA medium and colony diameters were measured every 24 h. (**C**) Colony morphology of WT, Δ*SsNEP2*, and *SsNEP2-C* strains after 14 days. (**D**) Number of sclerotia per plate. (**E**) Sclerotial weight per plate. WT, wild-type strain; Δ*SsNEP2*, knockout strain; *SsNEP2-C*, complementation strain. Experiments were conducted three times. Error bars indicate standard deviation (SD). Statistical significance was analyzed using Student’s *t*-test between wild-type and knockout mutant or complementation strain.

**Figure 3 pathogens-11-00446-f003:**
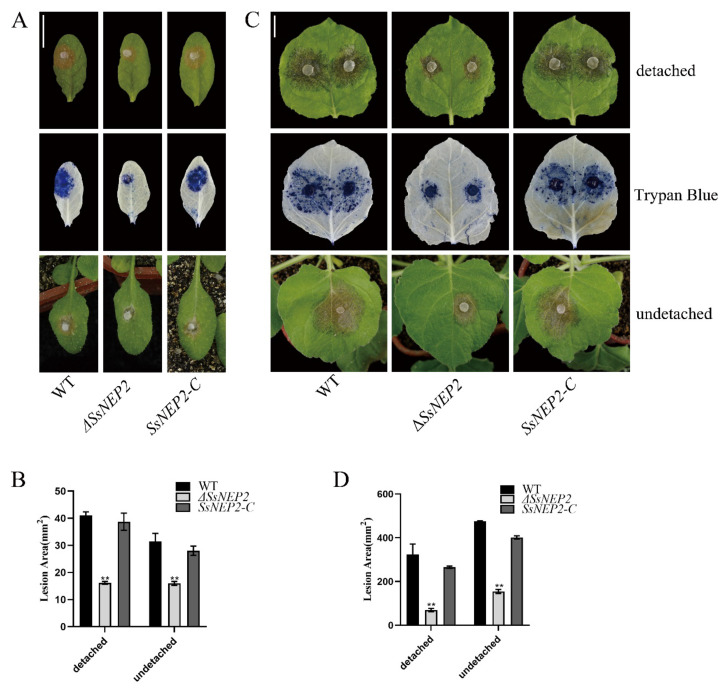
*SsNEP2* is associated with virulence. (**A**,**C**) Inoculated lesions and trypan blue staining of WT, *SsNEP2* deletion, and complementation strains of detached and undetached leaves of *A. thaliana* and *N. benthamiana*. Data were recorded at 24 hpi. Bar = 10 mm. (**B**,**D**) Lesion areas of WT, Δ*SsNEP2*, and SsNEP2-C on leaves of *A. thaliana* and *N. benthamiana*. WT, wild-type strain; Δ*SsNEP2*, knockout strain; *SsNEP2-C*, complementation strain. ImageJ was used to analyze lesion areas. Experiments were conducted three times. Error bars represent SD. Statistical significance was analyzed using Student’s *t*-test between wild-type and knockout mutant or complementation strain (** *p* < 0.01).

**Figure 4 pathogens-11-00446-f004:**
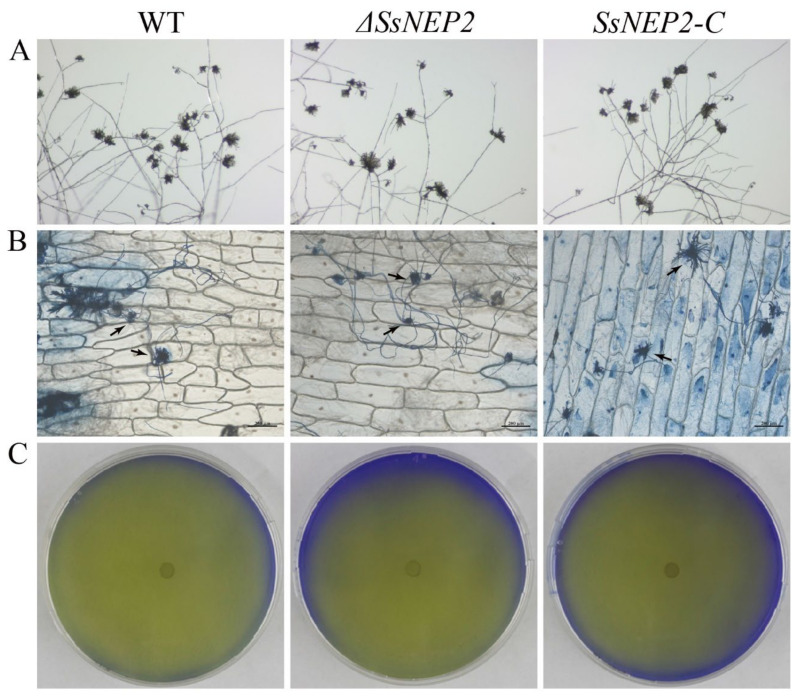
*SsNEP2* does not affect oxalate secretion and appressorium formation. (**A**) WT, Δ*SsNEP2*, and *SsNEP2-C* placed on glass slides and cultured for 16 h to observe formation and number of appressoria. (**B**) Penetration assay of WT, Δ*SsNEP2*, and *SsNEP2-C* on onion epidermis cells. Invasion hyphae were stained with trypan blue. Bar = 0.2 mm. (**C**) Mycelium of WT, Δ*SsNEP2*, and *SsNEP2-C* grown on PDA medium containing bromophenol blue. Experiments were conducted three times. WT, wild-type strain; Δ*SsNEP2*, knockout strain; *SsNEP2-C*, complementation strain.

**Figure 5 pathogens-11-00446-f005:**
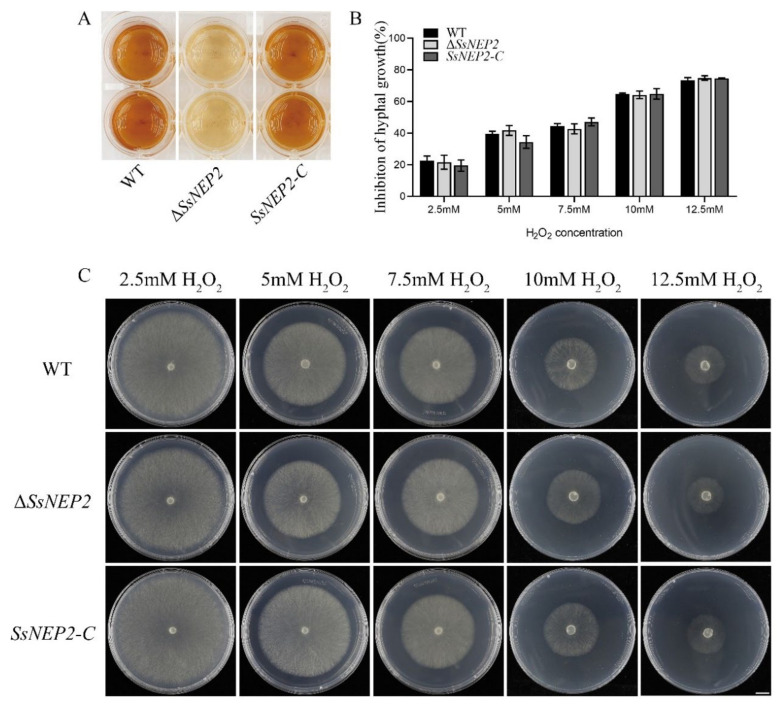
*SsNEP2* affects ROS accumulation. (**A**) DAB staining of WT, Δ*SsNEP2*, and *SsNEP2-C*. (**B**) Growth inhibition rate of WT, Δ*SsNEP2*, and *SsNEP2-C*. (**C**) Colonial morphology and mycelial growth of WT, Δ*SsNEP2*, and *SsNEP2-C* under different concentrations of H_2_O_2_. Bar = 10 mm. Experiments were conducted three times. WT, wild type; Δ*SsNEP2*, knockout strain; *SsNEP2-C*, complementation strain.

**Figure 6 pathogens-11-00446-f006:**
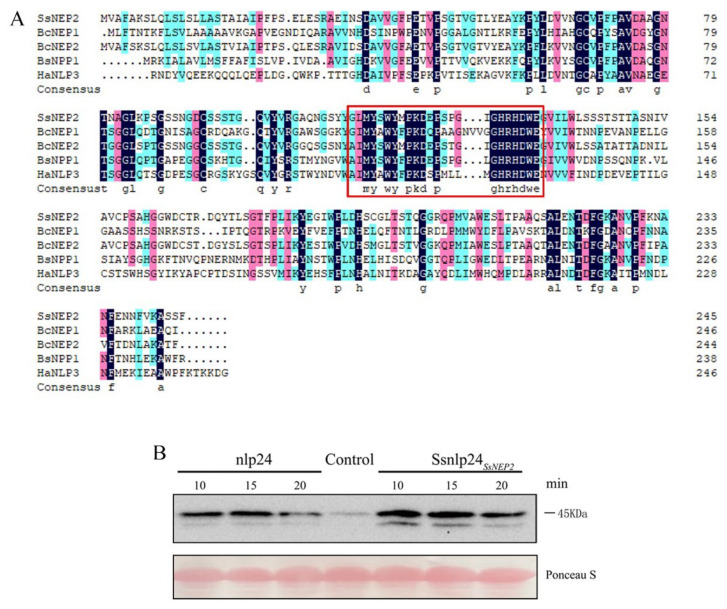
*Ss*nlp24 triggers MAPK activation in *Arabidopsis*. (**A**) Multiple sequence alignment of *Ss*NEP2 (SS1G_11912), *Bc*NEP1(XP_001555180), *Bc*NEP2(XP_001551049), *Bs*NPP1(WP_019714591.1). and *Ha*NLP3(6QBE_A). Red box: conserved nlp24 region. (**B**) *Ss*nlp24*_SsNEP2_*-induced MAPK activation. Seedlings were treated with 1 μM nlp24 or *Ss*nlp24*_SsNEP2_*. “Control” refers to the untreated “0” time point. Western blot with anti-pERK antibody is used for analysis of MAPK activation. Equal loading confirmed by Ponceau staining of Rubisco. Experiments were conducted three times.

**Figure 7 pathogens-11-00446-f007:**
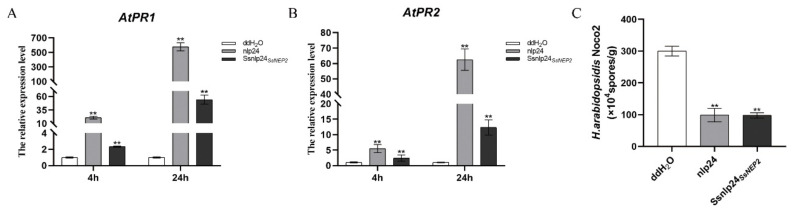
*Ss*nlp24*_SsNEP2_* enhances plant immunity. (**A**,**B**) Relative expression levels of *PR1* and *PR2* in Col-0 after treatment with 1 μM nlp24 or *Ss*nlp24*_SsNEP2_* for 4 or 24 h. Error bars represent SD. Statistical significance was analyzed using Student’s *t-*test between wild-type and knockout mutant or complementary strain (** *p* < 0.01). (**C**) Three-week-old soil-grown plants of indicated genotypes were pretreated with H_2_O and 1 μM nlp20 or *Ss*nlp24*_SsNEP2_* and sprayed with *H. a.* Noco2 spores (30,000 spores/mL) 24 h later. Experiments were conducted three times. Error bars represent SD. Statistical significance was analyzed using Student’s *t-*test between ddH_2_O, nlp24 and *Ss*nlp24*_SsNEP2_* (** *p* < 0.01).

## Data Availability

Not applicable.

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
