# Peer review of "SsNEP2 Contributes to the Virulence of Sclerotinia sclerotiorum"

_pathogens, 2022, doi:10.3390/pathogens11040446_

Round 1
Reviewer 1 Report
The current manuscript aims at assessing the role of SsNEP2 in the virulence of S. sclerotinia by using strains carrying the NEP2 deletion and exposing plant cells to the host immunity triggering core peptide (nlp24) of SsNEP2.
Although, based on leaf infection tests, the ∆ssNEP2 isolate exhibit significantly reduced virulence; in the end, besides less ROS production, we do not have a clear conclusion on how SsNEP2 may impact virulence of the pathogen (indirect metabolite-based or proteinaceous effector). Activating MAPK cascades and increasing resistance via the PTI pathways is linked to provoking the host environment, as an avirulence factor, to develop control over the invading pathogen.
To identify the functions and possible specialization of the NEP genes, the combined comparative analysis of both NEP1 and NEP2 genes should have been considered using their double and single deletions!!
However, the presented data might still deserve publication. The current data needs a disciplined introduction and a more elaborate discussion, based on additional references, by considering the phytotoxicity and disease susceptibility issues of various NLP proteins in Arabidopsis. (I suggest Chen et al. New Phytologist 2021; Costa Arenas et al. PMPP, 2010).
The title of the ms must also be changed according to the final relevant conclusions of the data
Further minor comments
- The “second” Figure 3 and Figure 4 must be correctly numbered (4 and 5) in the text.
- Figure 6/b – please indicate in the figure legend that “Control” refers to the untreated “0” timepoint. You might also consider restructuring the figure, placing the 0 timepoints ahead of the time courses.
Author Response
We genuinely thank you for your constructive and helpful suggestions and comments, which help us to improve the quality and thoroughness of our study. The manuscript texts have since been revised substantially. Some minor mistakes and typos were also corrected during revision. Our response in red font can be found beneath each original reviewer’s comment below.

Reviewer 2 Report
In this study, the authors attempted to clarify the molecular mechanism of pathogen virulence through investigating functions of the SsNEP2 in Sclerotinia sclerotiorum. Authors designed a series of experiments to test the hypothesis; however, lacking a comprehensive literature review and discussion, unjustified objectives and research design, and misinterpretation of the results diminished their efforts.
First, authors state that this study aims to investigate the functions of necrosis and ethylene-inducible peptide 2 (SsNEP2) in S. sclerotiorum (Line 12). However, literature review and discussion are all about NEPs. If the authors mutated the SsNEP2, the whole context is irrelevant with the topics. This part required authors’ clarification.
Below I will explain the major flaws of each section that need to be corrected before the manuscript is accepted. The review was done under the assumption that authors mistakenly named the gene of interest as SsNEP2 instead of SsNLP2.
Introduction
The introduction did not provide sufficient background information to make the article intelligible to readers in other disciplines. Lots of relevant information, which should be used for identifying the knowledge gap and justifying objectives, are either missing or underrepresented in the context. For example, 1) a comprehensive review regarding the known functions of NEPs in other organisms is needed. 2). Information about host defense mechanisms against fungal pathogens, especially Sclerotinia sclerotiorum, is lacking. 3) Known virulence factors (of S. sclerotiorum) that are related to this study are not well-reviewed, such as oxalic acid and its role in regulating ambient pH to facilitate the infection process. Additionally, lots of statements require citations that are missing (please see edited manuscripts).
Results
Authors only presented data generated from a single mutant raising my concern. From the results of generating gene-deletion mutant (Fig. 1B), it gives me the impression that authors have generated at least two mutants. Yet only one mutant was used for the subsequent experiments. Authors need to clarify that if they perform the experiments with both mutants and these two mutants show a similar phenotype. If not, the authors need to provide a compelling reason for using one mutant.
The word “pathogenicity” is used interchangeably with “virulence” throughout the manuscript. From my understanding, pathogenicity is a qualitative term used for describing the microbe is either pathogenic or not. On the other hand, virulence is quantitative, meaning there are degrees of disease severity. Therefore, these two terms are not interchangeable. SsNEP2-deletion mutant still can cause lesions on the leaves (Fig. 3), so the mutant should be referred to as a “low virulent” strain instead of a “low pathogenic” strain.
Experimental design for testing the hypothesis “Knockout of SsNEP2 affects ROS accumulation” is questionable. First, authors failed to provide a significant relevance between ROS production and NEP. Second, author did not disclose the correlation between ROS production and oxidation response. The data that the author presented here (Fig. 5) does not support the claim “It suggested that SsNEP2 protein does not affect the response of S. sclerotiorum to oxidative stress and may not interact with catalase”. The first part is supported by the results, but the second part requires additional experiments to verify the NEP2-catalase interaction. Furthermore, the connection between NLPs and catalase lacks support from the literature.
In Fig. 6C, authors attempt to testify the interaction among nlp24, AtRLP23, AtSOBIR1, and AtBAK1 using qRT-PCR, but I doubted that you can verify the protein-protein interactions using qRT-PCR assay alone. Related questions are:
- Does nlp24-AtRLP23 interaction lead to upregulation of AtRLP23 expression?
- Have you quantified ethylene levels in the plant after Ssnlp24SsNEP2 treatment, which provide strong evidence that Ssnlp24SsNEP2 triggers plant immunity mediated by RLP23–SOBIR1–BAK1 complex?
- Does the perception of nlp24 lead to upregulation of RLP23, SOBIR1, and BAK1-coding genes?
Discussion
Overall, authors mostly repeated the conclusion of each experiment stated in the Result. Authors need to provide an in-depth discussion about how SsNEP2 involves in fungal virulence and the role of Ssnlp24SsNEP2 in RLP23–SOBIR1–BAK1 complex-mediated immunity. Furthermore, the gap of the knowledge filled by this study is not well stated.
It has been demonstrated that AtRLP23 is required for nlp20 or nlp24-induced resistance. Treatment of nlp20 or nlp24 enhances Arabidopsis resistance to Botrytis cinerea infection. Additionally, ectopic expression of AtRLP23 in potato transgenic line also increases plant resistance to S. sclerotiorum (ref: http://dx.doi.org/10.1038/nplants.2015.140). According to this research, SsNEP2-deletion mutant does not synthesize SsNEP2 containing Ssnlp24SsNEP2, which serves as a PAMP and triggers plant PR genes expression as shown in Fig. 7A & B. However, this finding raises a question: why the SsNEP2-deletion mutant becomes a low virulent strain when it lost Ssnlp24SsNEP2? Any possible explanation?
Some misinterpretations of the results were found in the discussion:
- Line 269-276 Authors claimed that “It is suggested that SsNEP2 may affect the virulence of S. sclerotiorum through regulating the concentration of ROS” based on Fig. 5A. However, the results (Fig. 5A) only demonstrate that the level of cellular ROS is lower in the mutant. Authors need to evaluate the ROS level in the plant before making that speculation.
- Line 271-276 Authors stated that “the performance of ΔSsNEP2 was not different from that of the wild type, indicating that SsNEP2 does not affect the clearance of H2O2 in S. sclerotiorum”. Yet the results (Fig. 5c) merely show that deletion of SsNEP2 does not affect S. sclerotiorum tolerates H2O2. Authors need to design other experiments to address if SsNEP2 controls H2O2 clearance ability in S. sclerotiorum.
Materials and Methods
This section is extremely incomplete. Lots of information is scattered in the results, figure legends, and discussion. Many methods are not included in this manuscript, for example, plant materials and growth conditions, plant cell viability test, and statistical analysis. More comments and minor editing can be found in the attached edited manuscript.

Author Response

(The authors gave the same response as above.)

Round 2
Author Response
We genuinely thank you again for your constructive and helpful suggestions and comments, which help us to improve the quality and thoroughness of our study. The manuscript texts have since been revised substantially. Some minor mistakes and typos were also corrected during revision. Our response in red font can be found beneath each original reviewer’s comment below.

Reviewer 2 Report
Authors have addressed most of the comments and suggestions, however, some parts still have not been corrected. Please find my comments attached.

Author Response

(The authors gave the same response as above.)
